# Rainbow Trout (*Oncorhynchus mykiss*) Spleen-Derived Bioactive Compounds Suppress Pro-Inflammatory Gene Networks via NF-κB Pathway Modulation

**DOI:** 10.3390/genes16070837

**Published:** 2025-07-18

**Authors:** Do-Yeon Kim, Woo-Sung Choi, Ju-Hee Park, Seoghyun Kim, Jinyoung Park, Woohyun Song, Heejung Yang, Han-Heom Na, Keun-Cheol Kim

**Affiliations:** 1Department of Biological Sciences, College of Natural Sciences, Kangwon National University, Chuncheon 24341, Republic of Korea; dodo_96@kangwon.ac.kr (D.-Y.K.); 202516282@kangwon.ac.kr (W.-S.C.); pjhee@kangwon.ac.kr (J.-H.P.); seoghyunkim@kangwon.ac.kr (S.K.); hanhum01@kangwon.ac.kr (H.-H.N.); 2Laboratory of Natural Products Chemistry, College of Pharmacy, Kangwon National University, Chuncheon 24341, Republic of Korea; jypark@kangwon.ac.kr (J.P.); 202416113@kangwon.ac.kr (W.S.); heejyang@kangwon.ac.kr (H.Y.)

**Keywords:** rainbow trout (*Oncorhynchus mykiss*), spleen crude extract, fractionation, inflammation, NF-κB signaling pathway

## Abstract

**Background**: Natural products are key sources of anti-inflammatory agents, yet the potential of fish visceral extracts remains largely unexplored. This study evaluated the anti-inflammatory activity of a spleen extract from rainbow trout (*Oncorhynchus mykiss*). **Methods**: A crude spleen extract and its four solvent fractions were tested in LPS-stimulated RAW264.7 macrophages. Nitric oxide production and expression of iNOS, COX-2, and cytokines were assessed by qRT-PCR and Western blotting. The most active fraction, OSB (n-butanol layer), was further analyzed for its effects on NF-κB signaling, macrophage polarization, and ROS generation. **Results**: The crude spleen extract significantly reduced NO production and downregulated iNOS and COX-2 expression at both the transcriptional and translational levels. Among the four fractions, the OSB fraction exhibited the most potent and consistent anti-inflammatory effects. OSB markedly suppressed LPS-induced expression of iNOS, COX-2, and pro-inflammatory cytokines, while enhancing anti-inflammatory cytokines. Mechanistic analyses demonstrated that OSB inhibited NF-κB activation by preventing the nuclear translocation of the p65 subunit. Additionally, OSB attenuated LPS-induced ROS production and reduced the expression of M1 macrophage markers, indicating inhibition of M1 polarization. **Conclusions**: The OSB fraction from rainbow trout spleen exhibits potent anti-inflammatory activity by modulating the NF-κB pathway and suppressing M1 macrophage polarization, suggesting its potential as a natural therapeutic agent.

## 1. Introduction

Inflammation is a crucial component of the innate immune response triggered by microbial infections, tissue injury, or antigenic invasion. It is typically characterized by symptoms such as pain, swelling, itching, and fever, which accompany the elimination of pathogens and damaged cells from affected tissues [1]. Although the inflammatory response is generally self-limiting, its dysregulation or persistence can lead to chronic inflammatory conditions and contribute to the development and progression of various diseases, including cancer, diabetes, inflammatory bowel disease, asthma, and other autoimmune disorders [2]. Conventional anti-inflammatory agents, which primarily act by modulating inflammation-related cytokine expression, remain the standard treatment for these conditions [3]. However, their prolonged use is frequently associated with adverse effects such as gastrointestinal ulceration, mucosal perforation, and renal toxicity. These limitations underscore the urgent need for safer and more effective therapeutic alternatives with a reduced risk of drug resistance [4,5].

Macrophages, as key effectors of the innate immune system, play a central role in orchestrating inflammatory responses via activation of the Toll-like receptor 4 (TLR4)/nuclear factor kappa-light-chain-enhancer of activated B cells (NF-κB) signaling pathway upon lipopolysaccharide (LPS) stimulation [6,7]. Dysregulation of this pathway is implicated in numerous inflammatory diseases, including rheumatoid arthritis, pneumonia, and inflammatory bowel disease, making it a critical therapeutic target [8,9,10,11].

In recent years, natural product-derived compounds have gained significant attention for their ability to modulate inflammation-related signaling pathways [12,13]. The development of anti-inflammatory agents from natural sources is considered a promising alternative to synthetic drugs due to their favorable safety profiles and cost-effectiveness [14,15]. Notably, approximately 25% of anti-inflammatory drugs approved by the U.S. Food and Drug Administration (FDA) are derived from natural products, highlighting their therapeutic potential [16]. Fish processing by-products, such as heads, viscera, and bones—often comprising up to 50% of the total body mass depending on the species—are typically discarded through landfilling or incineration, contributing to environmental pollution, including greenhouse gas emissions, water contamination, and ecosystem disruption [17,18,19]. Repurposing these by-products offers not only a sustainable waste management strategy, but also a valuable opportunity to discover novel bioactive compounds from marine sources [20].

Previous studies have reported anti-inflammatory effects from various fish by-products. For example, cutlassfish head peptone was shown to modulate the mitogen-activated protein kinase (MAPK) signaling pathway in macrophages [21], and extracts from olive flounder heads suppressed iNOS and COX-2 expression [22]. Similarly, salmon bone extracts inhibited nitric oxide production, suggesting anti-inflammatory potential [23]. Despite these findings, the anti-inflammatory properties of fish visceral organ extracts remain poorly characterized and underexplored.

Among fish species, rainbow trout (*O. mykiss*) has gained attention for its high content of polydeoxyribonucleotide (PDRN)—a DNA-derived compound with molecular weights ranging from 50 kDa to 1500 kDa [24]. PDRN has been shown to attenuate LPS-induced acute lung injury in rats by inhibiting both the NF-κB and MAPK pathways [25], and to promote wound healing through the upregulation of vascular endothelial growth factor (VEGF) in diabetic mouse models [26]. Additionally, PDRN enhances osteoblast proliferation [27], increases collagen synthesis in incisional wound models [28], and exerts anti-inflammatory effects by modulating cytokine expression in osteoarthritic cell systems [29]. These findings collectively highlight the therapeutic potential of PDRN in treating inflammatory conditions [30]. However, commercial PDRN production typically relies on sperm- or milt-derived extraction from rainbow trout, which is limited by high production costs and the seasonal availability of spermatozoa [31].

In this study, we aimed to elucidate the molecular mechanisms underlying the anti-inflammatory effects of visceral extracts derived from rainbow trout using an LPS-stimulated macrophage model. Our findings provide novel insights into the bioactivity of fish by-product-derived visceral organ extracts and their potential as sustainable sources of anti-inflammatory therapeutics.

## 2. Materials and Methods

### 2.1. Preparation of Crude Extract and Fractionation Samples

Thirteen-month-old female rainbow trout were obtained from Woori Trout Farm (Chuncheon, Gangwon-do, Republic of Korea). Following dissection, their liver, gallbladder, spleen, and kidneys were harvested and extracted with 70% ethanol at 60 °C for 16 h. The resulting crude extracts were lyophilized, reconstituted in 70% ethanol, and subsequently diluted in a culture medium for use in in vitro experiments. The crude spleen extract was suspended in distilled water and sequentially partitioned with *n*-hexane, ethyl acetate, and *n*-butanol to yield solvent-soluble fractions. Initially, *n*-hexane was added to the aqueous suspension to obtain the *n*-hexane-soluble fraction. The remaining aqueous layer was subsequently extracted with ethyl acetate and then with *n*-butanol to yield the respective ethyl acetate- and *n*-butanol-soluble fractions. The final residual aqueous layer was collected as the water-soluble fraction. As a result, the crude extract was separated into four distinct fractions.

### 2.2. Cell Culture

RAW264.7 murine macrophage cells were cultured at 37 °C in a humidified atmosphere containing 5% CO_2_. The cells were maintained in DMEM supplemented with 10% fetal bovine serum (FBS) and 1% penicillin-streptomycin (P/S). The cells were gently detached using a cell scraper without trypsinization. Freeze-dried extracts from the rainbow trout liver, gallbladder, spleen, and kidneys were dissolved in 70% ethanol and diluted in the culture medium to the desired concentrations. Lipopolysaccharide (LPS) was dissolved in triple-distilled water and diluted to a final concentration of 1 μg/mL in DMEM.

### 2.3. Nitric Oxide Assay

RAW264.7 cells were seeded in 96-well plates and incubated for 24 h. The cells were then treated with crude extracts from the liver, gallbladder, spleen, or kidneys of the rainbow trout in combination with 1 μg/mL LPS. RAW264.7 cells were treated with each extract at concentrations of 10, 20, 40, 50, 100, and 200 μg/mL. After 24 h, the culture supernatant was collected and transferred to a new 96-well plate for nitric oxide measurement using the NO Plus Detection Kit (iNtRON Biotechnology, Seongnam, Republic of Korea), according to the manufacturer’s protocol.

### 2.4. Quantitative Reverse Transcription PCR (qRT-PCR)

RAW264.7 cells were seeded in culture dishes and incubated for 24 h, followed by treatment with OSH, OSE, OSB, or OSW in the presence of 1 μg/mL LPS for an additional 24 h. Total RNA was isolated using TRIzol reagent (Invitrogen, Carlsbad, CA, USA), and cDNA synthesis was performed using oligodT primers and M-MLV reverse transcriptase (Promega, Madison, WI, USA). qRT-PCR was conducted using the TOPreal™ qPCR 2X PreMIX (Enzynomics, Daejeon, Republic of Korea). Relative gene expression levels were calculated using the 2^−ΔΔCt^ method, with normalization to GAPDH. The primer sequences are listed below (Table 1):

### 2.5. Western Blotting

RAW264.7 cells were seeded into 100 mm dishes and treated 24 h later according to experimental conditions. After treatment, the cells were harvested using a scraper, and total protein was extracted using a RIPA lysis buffer (10 mM Tris-HCl pH 8.0, 1 mM EDTA, 140 mM NaCl, 1% Triton X-100, 0.1% sodium deoxycholate, and 0.1% SDS) containing a cOmplete™ Protease Inhibitor Cocktail (Roche, Basel, Switzerland). Protein concentration was measured using the Bradford assay (Thermo Fisher Scientific, Waltham, MA, USA). Proteins were separated via SDS-PAGE and transferred to 0.45 μm PVDF membranes. The membranes were blocked with 5% skim milk for 30 min at room temperature and then incubated overnight at 4 °C with primary antibodies diluted in 1× TBST containing 1% BSA. After washing, the membranes were incubated with HRP-conjugated secondary antibodies diluted in 5% skim milk for 2 h at room temperature. Protein bands were visualized using an ECL detection kit (GE Healthcare, Chicago, IL, USA). iNOS (13120S), COX-2 (12282S), IκBα (9242S), p-p65 (3033S), and CD86 (19589S) antibodies were purchased from Cell Signaling Technology (Danvers, MA, USA), and p65 (ab32536) and Lamin B1 (ab16048) antibodies were obtained from Abcam (Cambridge, UK).

### 2.6. Immunofluorescence

RAW264.7 cells were seeded onto coverslips in 6-well plates and treated 24 h after seeding. The cells were fixed with 4% paraformaldehyde for 10 min, permeabilized with 0.1% Triton X-100 in PBS for 3 min, and blocked with 5% skim milk in PBS for 1 h at room temperature. The cells were incubated with primary antibodies (1:200 dilution) for 2 h, followed by Alexa Fluor 488-conjugated goat anti-mouse IgG (1:200; Abcam, Boston, MA, USA) for 1 h. Nuclei were stained with DAPI (1:2000 in PBS). Fluorescence images were acquired using a confocal microscope (Nikon, Tokyo, Japan).

### 2.7. Reactive Oxygen Species (ROS) Analysis

RAW264.7 cells were seeded in 60 mm dishes and treated according to experimental protocols 24 h after seeding. The cells were incubated with 10 μM carboxy-H_2_DCFDA (Invitrogen, Carlsbad, CA, USA) at 37 °C for 30 min. After washing twice with PBS, intracellular ROS levels were measured using a flow cytometer (FACSymphony, Bergen County, NJ, USA) with excitation at 488 nm and emission at 525 nm.

### 2.8. Statistical Analysis

Statistical analyses were performed using GraphPad Prism 8.0 and ImageJ v1.54d. Data are presented as the mean ± standard deviation (SD) from at least three independent experiments. Comparisons between two groups were evaluated using Student’s *t*-test. A *p*-value < 0.05 was considered statistically significant, with the following notations: *p* ≤ 0.05 (*) and *p* ≤ 0.01 (**).

## 3. Results

### 3.1. Anti-Inflammatory Effect of Rainbow Trout Spleen Crude Extract in LPS-Stimulated RAW264.7 Cells

To investigate the bioactivity of organ-derived extracts from rainbow trout, crude extracts were prepared from the liver, gallbladder, spleen, and kidneys using 70% ethanol (Figure 1A). The anti-inflammatory potential of each extract was evaluated by treating RAW264.7 macrophages with LPS in the presence of each crude extract. NO production—a key marker of inflammation—was measured to assess the inflammatory response. LPS stimulation induced a nearly 10-fold increase in NO production compared to that in the untreated controls. Importantly, none of the crude extracts alone triggered an inflammatory response. Among the organ extracts tested, only the spleen crude extract markedly reduced LPS-induced NO production in a concentration-dependent manner, whereas the liver, gallbladder, and kidney extracts had no significant effect (Figure 1B). To further examine the anti-inflammatory activity of the spleen extract, we analyzed the expression levels of inflammation-related genes. Quantitative real-time PCR demonstrated that the spleen extract significantly downregulated the LPS-induced expression of *iNOS* and *COX-2* mRNA in a dose-dependent manner (Figure 1C). Consistent with the mRNA findings, a Western blot analysis revealed that spleen extract treatment also reduced the protein expression levels of iNOS and COX-2 induced by LPS (Figure 1D). Collectively, these results indicate that the spleen crude extract of rainbow trout exerts potent anti-inflammatory effects by suppressing NO production and downregulating the expression of key pro-inflammatory mediators in LPS-stimulated RAW264.7 macrophages.

### 3.2. Fractionation of Rainbow Trout Spleen Crude Extract and Anti-Inflammatory Effects of Each Fraction

To identify the bioactive anti-inflammatory components within the spleen crude extract, Solvent partitioning based on polarity was employed to obtain four fractions: OSH (*O. mykiss* Spleen crude extract n-Hexane layer), OSE (*O. mykiss* Spleen crude extract Ethyl acetate layer), OSB (*O. mykiss* Spleen crude extract n-Butanol layer), and OSW (*O. mykiss* Spleen crude extract Water layer) (Figure 2A). The chemical composition of each fraction was analyzed through ultra-performance liquid chromatography-mass spectrometry (UPLC-MS), revealing distinct profiles among the fractions (Figure 2B). To evaluate and compare the anti-inflammatory activity of each fraction, RAW264.7 cells were treated with 1 μg/mL LPS in the presence of either the crude spleen extract or individual fractions (OSH, OSE, OSB, or OSW) at a concentration of 200 μg/mL for 24 h. A quantitative PCR analysis was performed to assess the mRNA expression of *iNOS*, *COX-2*, and the pro-inflammatory cytokines *IL-6* and *TNF-α*. LPS stimulation significantly increased the expression of all four genes. Co-treatment with the spleen crude extract, OSH, OSE, or OSB attenuated this upregulation to varying degrees. Among the tested fractions, OSH, OSE, and OSB exhibited stronger inhibitory effects on inflammatory gene expression than the unfractionated crude extract. However, microscopic examination revealed that OSH and OSE induced cytotoxic effects, as indicated by the presence of floating and detached cells. In contrast, OSB treatment significantly suppressed the LPS-induced expression of *iNOS*, *COX-2*, *IL-6*, and *TNF-α* without causing observable cytotoxicity (Figure 2C,D). These results suggest that the principal anti-inflammatory activity of the spleen crude extract is enriched in the OSB fraction, which exerts potent anti-inflammatory effects while maintaining cell viability.

### 3.3. Regulatory Effect of OSB on iNOS and COX-2 Expression

Among the spleen-derived fractions, OSB exhibited the most potent and consistent anti-inflammatory activity, suggesting that the key bioactive constituents are predominantly enriched in this fraction. To further elucidate its mechanism of action, we examined the effects of OSB on iNOS and COX-2 protein expression in LPS-stimulated RAW264.7 cells. A Western blot analysis revealed that OSB treatment markedly suppressed the LPS-induced upregulation of iNOS and COX-2 protein levels in a dose-dependent manner (Figure 3A). In addition, a time-course experiment demonstrated that OSB effectively inhibited the time-dependent induction of both proteins following LPS stimulation (Figure 3B). Consistent with these results, an immunofluorescence analysis showed a marked decrease in the cytoplasmic fluorescence signals for iNOS and COX-2 following OSB and LPS co-treatment. In contrast, the cells treated with LPS alone exhibited strong fluorescence intensity, indicating robust protein expression (Figure 3C,D). Therefore, these results demonstrate that OSB effectively attenuates LPS-induced inflammatory responses by suppressing the expression of key pro-inflammatory enzymes—namely, iNOS and COX-2—in RAW264.7 macrophages.

### 3.4. Regulatory Effect of OSB on the NF-κB Signaling Pathway

To investigate the mechanism by which OSB regulates inflammatory responses, we examined its effects on the NF-κB signaling pathway, a key upstream regulator of *iNOS* and *COX-2* expression that is activated through LPS stimulation. In RAW264.7 macrophages, OSB treatment inhibited the LPS-induced phosphorylation of IκBα, thereby preventing its degradation. This inhibition subsequently suppressed the phosphorylation of the NF-κB subunit p65, which is dependent on IκBα activation (Figure 4A). A Western blot analysis further demonstrated that the OSB-mediated suppression of p65 phosphorylation resulted in reduced nuclear translocation of p65 following LPS stimulation (Figure 4B). Supporting these findings, an immunofluorescence analysis showed that p65 was translocated to the nucleus upon LPS stimulation, while OSB co-treatment blocked this translocation (Figure 4C). These results indicate that OSB attenuates LPS-induced inflammatory signaling by modulating the NF-κB pathway, thereby inhibiting the activation and nuclear translocation of p65.

### 3.5. Regulatory Effect of OSB on M1 Macrophage Polarization

The effects of OSB on macrophage polarization were assessed by analyzing the mRNA expression levels of pro- (*IL-6*, *TNF-α*, *IL-23p19*, and *IL-12p40*) and anti-inflammatory (*IL-13* and *TGF-β*) cytokines using quantitative PCR. OSB treatment significantly suppressed the LPS-induced upregulation of the pro-inflammatory cytokines, while simultaneously enhancing the expression of the anti-inflammatory cytokines *IL-13* and *TGF-β*. Furthermore, the expression of the M1 macrophage surface markers *CD40* and *CD86* was evaluated. LPS-induced increases in *CD40* and *CD86* mRNA levels were notably attenuated by the OSB treatment (Figure 5A). Protein-level changes in CD86 expression were confirmed through immunofluorescence staining, which demonstrated marked upregulation of CD86 on RAW264.7 cell surfaces following LPS stimulation; this increase was effectively inhibited by OSB (Figure 5B). Additionally, intracellular reactive oxygen species (ROS) levels were measured using the fluorescent probe H_2_DCFDA. In the graph, the x-axis represents fluorescence intensity detected in the FITC channel, displayed on a logarithmic scale. This intensity corresponds to the relative intracellular ROS levels, with higher values indicating greater ROS accumulation. The y-axis indicates the number of cells falling within each fluorescence intensity bin. OSB treatment dose-dependently reduced LPS-induced ROS production in RAW264.7 cells (Figure 5C). Collectively, these results indicate that OSB effectively suppresses LPS-induced M1 macrophage polarization by modulating cytokine expression, surface marker levels, and oxidative stress.

## 4. Discussion

The development of therapeutics derived from natural products for inflammatory diseases has garnered increasing attention in recent years [32,33,34]. A critical aspect of anti-inflammatory drug development involves elucidating the regulatory mechanisms that govern inflammation-associated signaling pathways [35]. Among these pathways, the TLR4/NF-κB pathway has been identified as a central target in the treatment of inflammatory disorders [36]. While several studies have explored the anti-inflammatory potential of fish by-product extracts, including those from the heads, bones, and fins, the biological activity of viscera-derived extracts remains largely underexplored [21,22,23]. Notably, extracts from the viscera of marine organisms such as *Turbo cornutus* and abalone have demonstrated anti-inflammatory activity by suppressing MAPK signaling components, including JNK and p38 phosphorylation [35,37].

Rainbow trout viscera have traditionally been considered industrial waste; however, recent efforts have focused on repurposing these by-products as valuable sources of bioactive compounds [38]. In this study, we investigated the anti-inflammatory effects of crude extracts from the liver, gallbladder, spleen, and kidneys of rainbow trout, with a focus on identifying active constituents with therapeutic potential. The initial evaluation was conducted via NO assays, as macrophages produce NO as a hallmark of LPS-induced inflammatory responses. Among the organ extracts tested, the spleen extract exhibited a dose-dependent inhibition of NO production. Furthermore, this extract significantly reduced both the mRNA and protein expression levels of iNOS as well as COX-2, another key inflammatory mediator. These results suggest that the spleen extract exerts a more pronounced anti-inflammatory effect relative to other tissue-derived extracts.

To isolate the bioactive constituents responsible for these effects, the spleen crude extract underwent solvent fractionation via liquid–liquid extraction based on differential polarity. This process yielded four fractions: OSH, OSE, OSB, and OSW. UPLC-MS analysis revealed distinct differences in the abundance of major components among the spleen crude extract and its fractions (OSH, OSE, OSB, and OSW). These findings indicate that the crude extract was effectively fractionated based on polarity, resulting in compositional variations across the different fractions. Each fraction, along with the crude extract, was evaluated for its ability to modulate the mRNA expression of inflammatory mediators (*iNOS*, *COX-2*, *IL-6*, and *TNF-α*). The OSH, OSE, and OSB fractions exhibited greater inhibitory activity than the crude extract. However, cytotoxic effects, including cell detachment and lysis, were observed following treatment with OSH and OSE. In contrast, the OSB fraction demonstrated strong and consistent anti-inflammatory activity without cytotoxicity, suggesting that it contains enriched bioactive compounds from the rainbow trout spleen.

A subsequent analysis confirmed that OSB treatment suppressed the LPS-induced expression of iNOS and COX-2 in a dose-dependent manner at the protein level. Time-course experiments revealed that although OSB suppressed these inflammatory proteins, their levels gradually increased over time, indicating partial rather than complete inhibition. These effects were validated through both Western blotting and immunofluorescence microscopy. Notably, OSB treatment reduced the cytoplasmic fluorescence intensity of iNOS and COX-2 induced by LPS, consistent with the biochemical results. These results align with the reported anti-inflammatory effects of natural compounds, such as resveratrol and quercetin [39,40], which similarly downregulate iNOS and COX-2 expression. Taken together, our findings provide mechanistic evidence supporting the anti-inflammatory activity of the OSB fraction.

Given the pivotal role of NF-κB signaling in regulating the transcription of iNOS, COX-2, and other pro-inflammatory genes, we further investigated the upstream mechanisms modulated by OSB. LPS stimulation leads to IκBα phosphorylation and degradation, facilitating the nuclear translocation of the NF-κB p65 subunit. OSB treatment inhibited both IκBα phosphorylation/degradation and p65 activation, thereby preventing its translocation to the nucleus. Western blotting and immunofluorescence analyses confirmed this suppression, demonstrating a reduction in nuclear p65 localization upon OSB treatment. Notably, this mode of action mirrors that of curcumin—a well-characterized natural compound known to suppress NF-κB activation [41]. Collectively, these findings suggest that OSB exerts its anti-inflammatory effects by preventing NF-κB activation and nuclear translocation, thereby downregulating the transcription of pro-inflammatory mediators.

Macrophage polarization is another critical determinant of inflammatory responses [42]. Upon LPS stimulation, macrophages differentiate into the pro-inflammatory M1 phenotype, characterized by the increased expression of surface markers (CD40 and CD86), the elevated production of pro-inflammatory cytokines, and enhanced ROS generation [43]. Our results demonstrated that OSB significantly downregulated the LPS-induced mRNA expression of IL-6, TNF-α, IL-12p40, and IL-23p19. In parallel, it upregulated anti-inflammatory cytokines, such as IL-13 and TGF-β. OSB also suppressed the expression of M1 surface markers CD40 and CD86 at both the mRNA and protein levels. The immunofluorescence analysis further confirmed reduced CD86 surface expression. Moreover, OSB effectively diminished LPS-induced ROS generation, suggesting inhibition of oxidative stress—a hallmark of M1 polarization. These results collectively support the notion that OSB suppresses M1 polarization in LPS-stimulated RAW264.7 macrophages.

This study evaluated the anti-inflammatory activity of the OSB fraction under in vitro conditions. Although OSB effectively modulated *iNOS* and *COX-2* expression through inhibition of the NF-κB pathway, the in vitro nature of this work limits direct conclusions regarding its therapeutic efficacy in complex physiological systems. Therefore, future studies employing in vivo animal models are necessary to assess the anti-inflammatory potency and safety profile of OSB under biologically relevant conditions. Moreover, OSB remains a crude extract, and the relatively high concentrations used in this study (up to 400 μg/mL) may not be feasible for clinical application. Accordingly, further work is required to chemically characterize the OSB fraction and isolate its active constituents. The identification of these bioactive components will be essential for elucidating the mechanisms of action and evaluating their pharmacological relevance in vivo.

Nevertheless, the anti-inflammatory properties demonstrated by OSB in this study are consistent with those reported for natural compounds such as resveratrol, quercetin, and curcumin [39,40,41]. These findings support the potential of OSB as a promising bioactive candidate and provide a strong rationale for further investigation.

## 5. Conclusions

This study demonstrated that the OSB fraction, derived from the spleen extract of rainbow trout (*O. mykiss*), exhibits potent anti-inflammatory activity without inducing cytotoxic effects. OSB effectively attenuated LPS-induced M1 macrophage polarization by downregulating the expression of *iNOS*, *COX-2*, pro-inflammatory cytokines, M1 surface markers, and intracellular ROS levels, primarily via inhibition of the NF-κB signaling pathway. However, as the current findings are based on in vitro experiments, further in vivo studies are required to validate the therapeutic efficacy and safety of OSB in physiological contexts. Additionally, the chemical characterization of OSB is necessary to identify its active constituents and to further elucidate their mechanisms of action. Collectively, these results provide mechanistic insight into the anti-inflammatory potential of fish viscera-derived bioactive compounds and suggest that the OSB fraction may serve as a promising candidate for the development of natural anti-inflammatory therapeutics.

## Figures and Tables

**Figure 1 genes-16-00837-f001:**
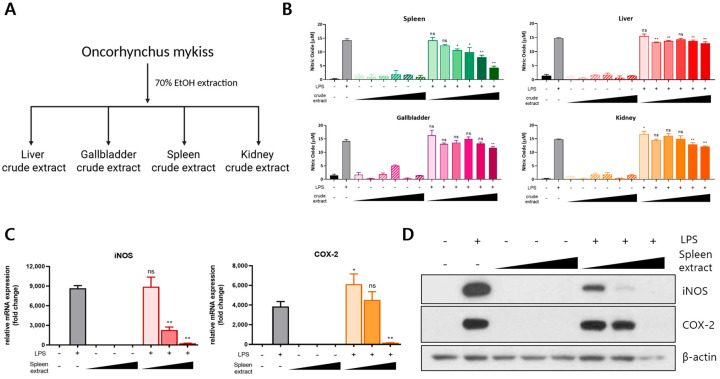
Anti-inflammatory effects of rainbow trout spleen crude extract in LPS-stimulated RAW264.7 cells. (**A**) Crude extracts were prepared from the liver, gallbladder, spleen, and kidneys of rainbow trout via 70% ethanol extraction at 60 °C for 16 h. (**B**) RAW264.7 cells were treated with each extract (10–200 μg/mL) in the presence of 1 μg/mL LPS, and nitric oxide (NO) production was quantified. (**C**) The mRNA expression levels of iNOS and COX-2 were assessed via qRT-PCR following treatment with the spleen crude extract (100–400 μg/mL). (**D**) The protein expression of iNOS and COX-2 was analyzed through Western blotting under the same treatment conditions. All treatments were performed for 24 h. The data are presented as mean ± SD (*n* = 3). A statistical analysis was conducted using one-way ANOVA followed by multiple comparisons. *p* ≤ 0.05 (*) and *p* ≤ 0.01 (**). ns: not significant.

**Figure 2 genes-16-00837-f002:**
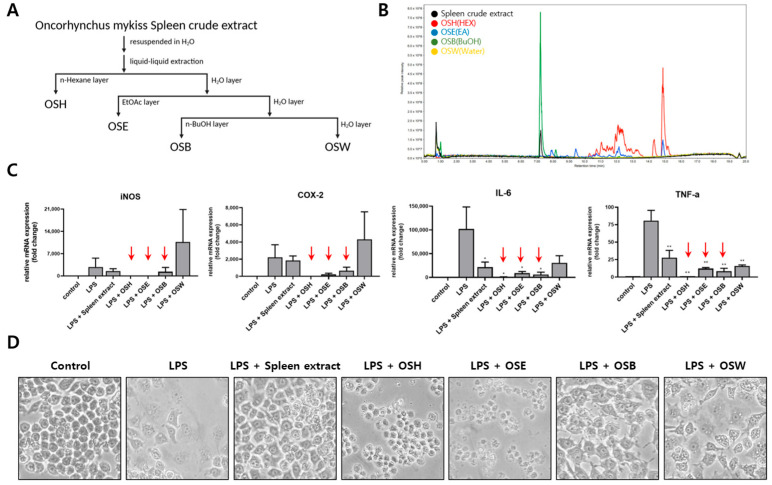
Fractionation of rainbow trout spleen extract and evaluation of anti-inflammatory activity of each fraction. (**A**) The spleen crude extract was fractionated into four solvent layers (OSH, OSE, OSB, and OSW) through polarity-based liquid–liquid extraction and lyophilized. (**B**) Major constituents of each fraction were characterized via a UPLC-MS analysis. (**C**) RAW264.7 cells were treated with the crude extract or each fraction (200 μg/mL) in the presence of 1 μg/mL LPS, and the mRNA expression of iNOS, COX-2, IL-6, and TNF-α was quantified through qRT-PCR. The red arrows indicate the fractions that showed stronger anti-inflammatory effects than the crude extract. (**D**) Morphological changes in the RAW264.7 cells were observed via phase-contrast microscopy following treatment. The data are presented as mean ± SD (*n* = 3). Statistical significance was determined using one-way ANOVA with multiple comparisons. *p* ≤ 0.05 (*) and *p* ≤ 0.01 (**).

**Figure 3 genes-16-00837-f003:**
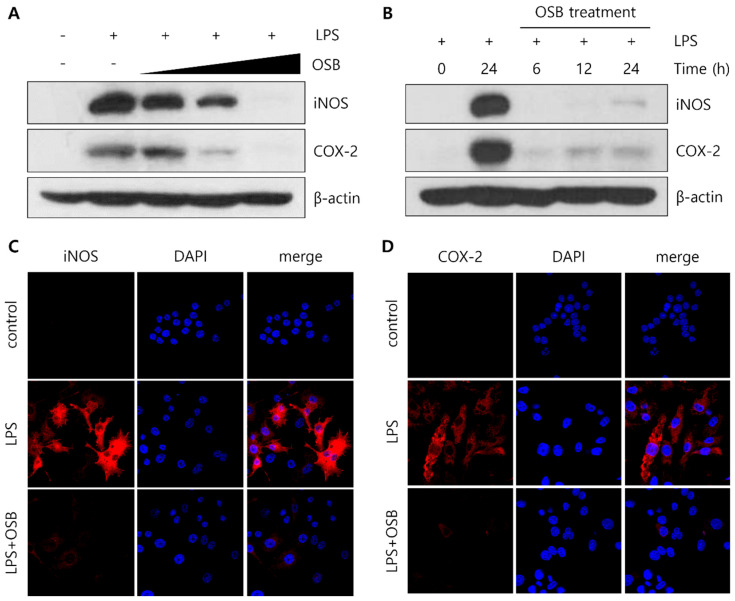
Regulatory effects of the OSB fraction on iNOS and COX-2 expression in LPS-stimulated RAW264.7 cells. (**A**) The dose-dependent effects of OSB (100–400 μg/mL) on iNOS and COX-2 protein expression were analyzed through Western blotting. (**B**) Time-course analysis of iNOS and COX-2 expression following treatment with OSB (200 μg/mL) for 6–24 h in the presence of LPS. (**C**,**D**) Immunofluorescence analysis of iNOS and COX-2 protein expression after treatment with OSB (200 μg/mL) for 24 h. All experiments were conducted with LPS stimulation (1 μg/mL, 24 h). In the immunofluorescence analysis, cell nuclei were stained with DAPI (blue), and iNOS and COX-2 were visualized with red fluorescence.

**Figure 4 genes-16-00837-f004:**
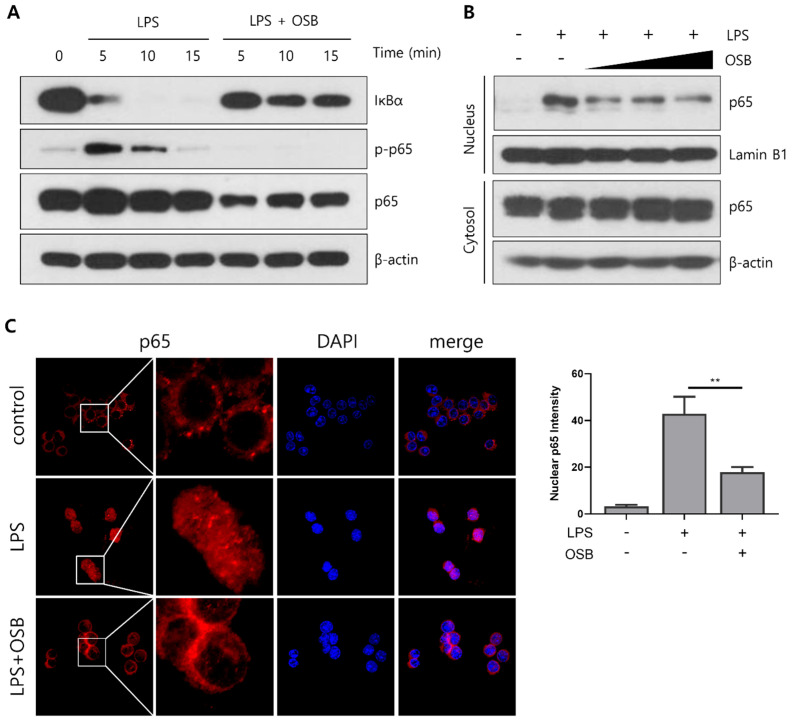
Inhibitory effects of OSB on NF-κB signaling pathway activation. (**A**) Western blot analysis of IκBα and p65 phosphorylation in RAW264.7 cells treated with OSB (200 μg/mL) in the presence of LPS. (**B**) The nuclear translocation of p65 was examined after treatment with OSB (100–400 μg/mL) for 2 h. (**C**) Immunofluorescence analysis of p65 localization following OSB treatment (400 μg/mL, 2 h). The nuclear p65 fluorescence intensity was quantified using ImageJ. All experiments included LPS stimulation (1 μg/mL). In the immunofluorescence analysis, cell nuclei were stained with DAPI (blue), and p65 was visualized with red fluorescence. *p* ≤ 0.01 (**).

**Figure 5 genes-16-00837-f005:**
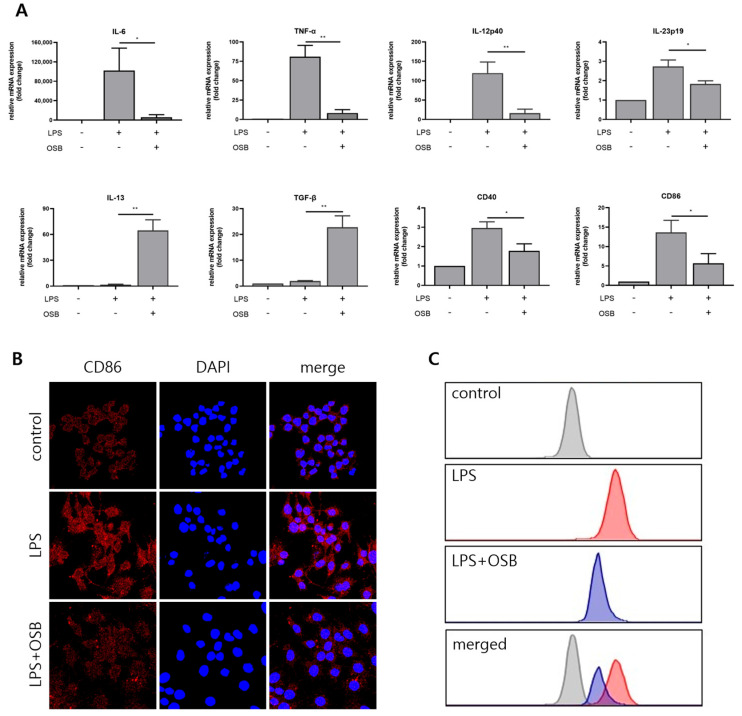
Effects of OSB on LPS-induced M1 macrophage polarization. (**A**) qRT-PCR analysis of pro-inflammatory cytokines (IL-6, TNF-α, IL-12p40, and IL-23p19), anti-inflammatory cytokines (IL-13 and TGF-β), and M1 surface markers (CD40 and CD86) following OSB treatment (200 μg/mL, 24 h). (**B**) Immunofluorescence analysis of CD86 expression in RAW264.7 cells treated with OSB (200 μg/mL). In the immunofluorescence analysis, cell nuclei were stained with DAPI (blue), and CD86 was visualized with red fluorescence. (**C**) The intracellular reactive oxygen species (ROS) levels were measured using H_2_DCFDA after OSB treatment (400 μg/mL, 24 h). All cells were stimulated with 1 μg/mL LPS. The data are expressed as mean ± SD (*n* = 3), and statistical significance was assessed through one-way ANOVA with multiple comparisons. *p* ≤ 0.05 (*) and *p* ≤ 0.01 (**).

**Table 1 genes-16-00837-t001:** qRT-PCR Primer Sequences.

Gene	The Sequences of the Primers
*iNOS*	F-CCAAGCCCTCACCTACTTCC
	R-CTCTGAGGGCTGACACAAGG
*COX-2*	F-CATCCCCTTCCTGCGAAGTT
	R-CATGGGAGTTGGGCAGTCAT
*IL-6*	F-AGTCCTTCCTACCCCAATTTCC
	R-TAACGCACTAGGTTTGCCGA
*TNF-* *α*	F-ACCGTCAGCCGATTTGCTAT
	R-TTGGGCAGATTGACCTCAGC
*IL-12p40*	F-AGACCCTGCCCATTGAACTG
	R-CAGGAGTCAGGGTACTCCCA
*IL-23p19*	F-CAGCAGCTCTCTCGGAATCT
	R-CAGACCTTGGCGGATCCTTT
*IL-13*	F-GTATGGAGTGTGGACCTGGC
	R-ATTTTGGTATCGGGGAGGCTG
*TGF-* *β*	F-CTGCTGACCCCCACTGATAC
	R-GGGGCTGATCCCGTTGATTT
*CD40*	F-GCTATGGGGCTGCTTGTTGA
	R-GGTGGCATTGGGTCTTCTCA
*CD86*	F-ATGGACCCCAGATGCACCA
	R-TGTGCCCAAATAGTGCTCGT
*GAPDH*	F-CTCATGACCACAGTCCATGC
	R-CACATTGGGGGTAGGAACAC

## Data Availability

The data that support the findings of this study are available from the corresponding author upon reasonable request.

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
