# Peer review of "Rainbow Trout (Oncorhynchus mykiss) Spleen-Derived Bioactive Compounds Suppress Pro-Inflammatory Gene Networks via NF-κB Pathway Modulation"

_genes, 2025, doi:10.3390/genes16070837_

Round 1

Reviewer 1 Report

Comments and Suggestions for Authors

The article addresses an interesting subject, and I do not find any flaw in the experimental design of this work.  However I think adding more details to the Materials and Methods section could improve the reproducibility of the analyses described, and in general the accuracy of the article. Moreover, I strongly recommend deepening the Discussion section.

In general, verify that all the genes are written in italics.

Indicate in the Abstract which system was used to assess the anti-inflammatory activity (in vitro assays, type of cells).

Line 18: Specify the acronym OSB that is mentioned here for the first time.

Materials and Methods section: Indicate what concentrations of the extracts were used for each experiment.

Chapter 2.1:

  • Have you performed the extraction in replicates?
  • Have you verified any difference in the extracts among replicates?
  • How much extract was obtained from each organ used?
  • Have you tested other extraction techniques or solvents?

Line 114: Specify what these acronyms stand for.

Line 195-200: This paragraph should be moved to the Materials and Methods section.

Figure 3: In the figure caption I suggest indicating what the colors are referring to (same comment applies for Figures 4 and 5).

Figure 5A: Are the numbers in the axis of the IL-6 graph correct?

Figure 5C: What are the axes of the graphs?

Lines 325-329: The UPLC-MS results are not properly discussed.

Author Response

Comment) The English is fine and does not require any improvement.

Answer) We are grateful for the reviewer’s comment on the manuscript’s language quality. To ensure greater clarity and fluency, we have carefully revised the text with the assistance of professional English editing services.

Comment) The article addresses an interesting subject, and I do not find any flaw in the experimental design of this work. However I think adding more details to the Materials and Methods section could improve the reproducibility of the analyses described, and in general the accuracy of the article. Moreover, I strongly recommend deepening the Discussion section.

Answer) Thank you for your comment. We have added the drug treatment concentrations used in the NO assay to the Materials and Methods section and supplemented the description of the liquid–liquid extraction process to improve the accuracy of the manuscript. In addition, we have enhanced the Discussion section by including a statement on the limitations of this study and directions for future research in the final paragraph. A Conclusion section has also been added to improve the overall completeness and depth of the manuscript.

Comment) In general, verify that all the genes are written in italics.

Answer) Thank you for your comment. All gene names have been revised to italic font throughout the manuscript.

Comment) Indicate in the Abstract which system was used to assess the anti-inflammatory activity (in vitro assays, type of cells).

Answer) Thank you for your comment. We have revised the Abstract to clarify that the results were evaluated through analytical methods. Additionally, we have specified that RAW264.7 cells, a murine macrophage cell line, were used in the experiments.

Comment) Line 18: Specify the acronym OSB that is mentioned here for the first time.

Answer) Thank you for your comment. We have revised the manuscript to include a clear definition of the abbreviation "OSB" at its first mention in the text.

Comment) Materials and Methods section: Indicate what concentrations of the extracts were used for each experiment.

Answer) Thank you for your comment. (Figure 1B) The drug treatment concentrations used in the NO assay have been added to the Materials and Methods section. Since the treatment conditions for qRT-PCR and Western blot analyses differ depending on the specific experiment, the respective concentrations have been indicated in each figure legend.

Comment) Chapter 2.1:

  • Have you performed the extraction in replicates?
  • Have you verified any difference in the extracts among replicates?
  • How much extract was obtained from each organ used?
  • Have you tested other extraction techniques or solvents?

Answer) Thank you for your comment. The extraction process of the rainbow trout organ extracts was performed three times in total, and the crude extracts obtained from each organ in each batch consistently retained their respective bioactivities. In particular, all three batches of the spleen extract exhibited consistent anti-inflammatory effects. Comparing the two largest and most abundant organs obtained—liver and spleen, which was the focus of this study—we obtained approximately 25 g of liver crude extract from 1 kg of trout liver and about 22 g of spleen crude extract from 1 kg of trout spleen. Similar yields were obtained from the other organs as well. The choice of 70% ethanol (EtOH) as the extraction solvent was made for several reasons. Since the exact constituents of the natural extracts are unknown, a solvent capable of extracting a broad range of compounds from polar to non-polar was desired. Extraction with 70% EtOH is more effective at simultaneously extracting multiple compounds compared to using a single solvent such as water or absolute ethanol alone. Therefore, 70% EtOH was selected as the extraction solvent to maximize the diversity of extracted components.

Comment) Line 114: Specify what these acronyms stand for.

Answer) We appreciate the reviewer’s comment. We have added explanations for the abbreviations OSH, OSE, OSB, and OSW in the main text.

Comment) Line 195-200: This paragraph should be moved to the Materials and Methods section.

Answer) We thank the reviewer for the valuable comment. We have supplemented the Materials and Methods section with additional details regarding the fractionation process. This was intended to enhance the clarity and rationale of the experimental material preparation and study design, thereby improving understanding and reproducibility.

Comment) Figure 3: In the figure caption I suggest indicating what the colors are referring to (same comment applies for Figures 4 and 5).

Answer) We appreciate the reviewer’s comment. We have actively incorporated the suggestion by clarifying the meanings of the blue and red colors to enhance the understanding of the data. This explanation has been added to the figure legend.

Comment) Figure 5A: Are the numbers in the axis of the IL-6 graph correct?

Answer) We thank the reviewer for the comment. All qRT-PCR experiments conducted in this study were performed with at least three biological replicates. When macrophages are stimulated with LPS, the expression of inflammatory cytokines increases sharply. Among these, IL-6 is barely expressed under normal conditions, so its fold change values show a dramatic increase. However, this pronounced change has been consistently confirmed through repeated experiments.

Comment) Figure 5C: What are the axes of the graphs?

Answer) We appreciate the reviewer’s comment. In the graph, the x-axis represents the fluorescence intensity detected in the FITC channel, displayed on a logarithmic scale. This intensity reflects the relative intracellular ROS levels, with higher values indicating increased ROS. The y-axis shows the number of cells within each fluorescence intensity bin. We have added this explanation to the Results section to improve the accuracy of data interpretation. This clarification helps to further support the evidence for OSB’s inhibitory effect on macrophage M1 polarization.

Comment) Lines 325-329: The UPLC-MS results are not properly discussed.

Answer) We sincerely appreciate the reviewer’s valuable suggestion. Following the advice, we have supplemented the manuscript by adding a discussion on the significance of the major component analysis of the spleen crude extract and its fractions. This content has been incorporated into the Discussion section.

Reviewer 2 Report

Comments and Suggestions for Authors
  1. The authors have chosen the RAW264.7 murine macrophage cell line for this study. While this is a widely used in vitro model for evaluating macrophage-mediated inflammatory responses and NF-κB signaling, the rationale for selecting a murine rather than a fish-derived cell line (e.g., from Oncorhynchus mykiss) should be justified, particularly given the species-specific nature of immune signaling pathways.

  2. The experimental design would benefit from the inclusion of a positive control, such as a well-characterized anti-inflammatory compound (e.g., dexamethasone or curcumin), to validate the observed suppression of pro-inflammatory gene expression. This would strengthen the conclusions by providing a benchmark against an established NF-κB inhibitor.

  3. The study currently lacks an in vivo model, which raises concerns about the translatability of the findings. In vitro systems like RAW264.7 offer controlled environments, but they cannot replicate the complex physiological interactions present in a live organism. The authors should justify the omission of an animal model and discuss how this limitation affects the broader applicability of the results, particularly in the context of systemic immune regulation in fish.

  4. The observed modulation of NF-κB signaling and suppression of pro-inflammatory gene networks by rainbow trout spleen-derived bioactive compounds may have promising applications in aquaculture. These findings suggest potential for developing novel anti-inflammatory therapeutics or functional feed additives aimed at enhancing fish health and resilience to stress or infection. However, further studies, including in vivo validation and assessment of long-term safety and efficacy, are necessary before practical application can be considered.

Author Response

Comment) The English could be improved to more clearly express the research.

Answer) We sincerely thank the reviewer for pointing this out. To enhance clarity and readability, we have revised the manuscript with professional English editing.

Comment) The authors have chosen the RAW264.7 murine macrophage cell line for this study. While this is a widely used in vitro model for evaluating macrophage-mediated inflammatory responses and NF-κB signaling, the rationale for selecting a murine rather than a fish-derived cell line (e.g., from Oncorhynchus mykiss) should be justified, particularly given the species-specific nature of immune signaling pathways.

Answer) Thank you for this insightful comment. The primary objective of our study was to identify and characterize novel anti-inflammatory compounds from rainbow trout (Oncorhynchus mykiss) spleen extract. To evaluate the immunomodulatory activity of these compounds, we used RAW264.7 murine macrophages, a well-established and widely used model for analyzing macrophage-mediated inflammatory responses and NF-κB signaling. Although fish-derived cell lines would offer species specificity, the RAW264.7 model allows for robust, reproducible assessment of inflammatory mechanisms and is suitable for initial functional screening of natural product extracts.

Comment) The experimental design would benefit from the inclusion of a positive control, such as a well-characterized anti-inflammatory compound (e.g., dexamethasone or curcumin), to validate the observed suppression of pro-inflammatory gene expression. This would strengthen the conclusions by providing a benchmark against an established NF-κB inhibitor.

Answer) We appreciate the reviewer’s valuable suggestion. In response, we have included a discussion comparing our findings with those of well-characterized anti-inflammatory compounds such as curcumin. Specifically, we discuss how OSB’s effects on NF-κB signaling and cytokine suppression align with the mechanisms reported for curcumin (ref. 41). This comparison strengthens the relevance of our results, even in the absence of a direct positive control experiment.

Comment) The study currently lacks an in vivo model, which raises concerns about the translatability of the findings. In vitro systems like RAW264.7 offer controlled environments, but they cannot replicate the complex physiological interactions present in a live organism. The authors should justify the omission of an animal model and discuss how this limitation affects the broader applicability of the results, particularly in the context of systemic immune regulation in fish.

Answer) We fully agree with the reviewer that in vitro findings require in vivo validation for translational relevance. Although our current study was limited to an in vitro system, we emphasized the necessity of future in vivo studies in the Conclusion section. Additionally, we have expanded the Discussion to highlight that anti-inflammatory effects of OSB resemble those of natural products such as curcumin, resveratrol, and quercetin, thereby supporting its potential as a candidate for further preclinical development.

Comment) The observed modulation of NF-κB signaling and suppression of pro-inflammatory gene networks by rainbow trout spleen-derived bioactive compounds may have promising applications in aquaculture. These findings suggest potential for developing novel anti-inflammatory therapeutics or functional feed additives aimed at enhancing fish health and resilience to stress or infection. However, further studies, including in vivo validation and assessment of long-term safety and efficacy, are necessary before practical application can be considered.

Answer) We sincerely thank the reviewer for this insightful comment. The primary objective of our study was to identify and characterize novel anti-inflammatory compounds derived from the spleen extract of rainbow trout (Oncorhynchus mykiss). Furthermore, this work aimed to explore new natural product candidates that may contribute to the development of anti-inflammatory therapeutics for human application. Although the current study was limited to an in vitro model, we have emphasized the need for further in vivo validation in the Conclusion section. While in vivo studies are still warranted, we believe our findings provide a valuable foundation for the future development of fish-derived bioactive compounds with anti-inflammatory potential. Additionally, we have expanded the Discussion section to include a comparison between OSB and well-characterized natural compounds such as curcumin, resveratrol, and quercetin, further supporting OSB’s potential as a candidate for preclinical development.

Reviewer 3 Report

Comments and Suggestions for Authors

I have included my comments in a separate Word file.

I have carefully and with great interest reviewed this original research. I am happy to report that it is almost ready for publication and needs, in my opinion, minor, if any adjustments. I have arranged my comments into specific (line-by-line) and general, so as to help the authors in their work.

Specific Comments

Line 54 – reference 14 (Gautam et al., 2009). Recent developments in anti-inflammatory natural products, while correct is quite dated, being published over a decade before. While not wrong, I would consider adding a more recent reference here in addition to the existing reference (preferably from the last 2-3 years). For example, there is a paper titled “Review on Anti-Inflammatory Activity of Natural Products’’ from 2025 (this is just an example, please use whichever recent publication you deem most appropriate).

Line 81 – I am not sure if the limitations in commercial PDRN production are clearly mentioned in ref. 30 (Kim et al., 2020. Polydeoxyribonucleotide Activates Mitochondrial Biogenesis but Reduces MMP-1 Activity and Melanin Biosynthesis in Cultured Skin Cells). Perhaps you can consider adding a reference where this point is made more clearly and preferably mentioned in the abstract?

Lines 196-197 – there is here a sentence which is in bold, with no apparent reason; I suppose this is a typo.

Line 220 – There is an improper space after sentence (B); I assume this is an error of text formatting.

Line 354 – I think you should add a reference here to document the M1 polarisation consequences.

General Comments

  1. I think the paper could benefit from a dedicated Conclusions section – perhaps you can consider making the last paragraph (lines 363-370) a Conclusions sections, after adding a few more lines summarising your research?
  2. You mention a lot about the inflammatory pathways (e.g., NF-κΒ, COX-2, etc); I think the manuscript would benefit from a comparison of the effects described in your text with those of other natural products, such as pinosylvin or curcumin (indicative references for these are “Pinosylvin: A Multifunctional Stilbenoid with Antimicrobial, Antioxidant, and Anti-Inflammatory Potential’’ and “Curcumin Attenuates Acrolein-induced COX-2 Expression and Prostaglandin Production in Human Umbilical Vein Endothelial Cells’’ – these are just indicative to orient you towards the specific aspect mentioned; there is not need to cite these papers. You may choose whichever you think are better for your purposes).
  3. Perhaps you can consider mentioned potential applications of your findings in real clinical situations, for example in viral or bacterial infections; also, I think it would be of benefit to provide some directions for future research.

Finally, I wish to congratulate the authors on their research and wish them best of luck in their research endeavours. Regarding the comments I made for additions, I just think that they would make your paper more complete and would increase its value. Other than that, if you feel they conflict with your vision of your paper, please do not feel obliged to adhere to them.

Author Response

Comment) The English is fine and does not require any improvement.

Answer) We appreciate the reviewer’s positive comment regarding the quality of the English. To enhance clarity and readability, we have revised the manuscript with professional English editing.

Specific Comments

Comment) Line 54 – reference 14 (Gautam et al., 2009). Recent developments in anti-inflammatory natural products, while correct is quite dated, being published over a decade before. While not wrong, I would consider adding a more recent reference here in addition to the existing reference (preferably from the last 2-3 years). For example, there is a paper titled “Review on Anti-Inflammatory Activity of Natural Products’’ from 2025 (this is just an example, please use whichever recent publication you deem most appropriate).

Answer) Thank you for pointing this out. We have added a more recent reference on anti-inflammatory natural products published within the past 2–3 years and retained the original citation for completeness (now cited as ref. 15).

Comment) Line 81 – I am not sure if the limitations in commercial PDRN production are clearly mentioned in ref. 30 (Kim et al., 2020. Polydeoxyribonucleotide Activates Mitochondrial Biogenesis but Reduces MMP-1 Activity and Melanin Biosynthesis in Cultured Skin Cells). Perhaps you can consider adding a reference where this point is made more clearly and preferably mentioned in the abstract?

Answer) We agree with the reviewer’s observation. The previous reference (Kim et al., 2020) did not adequately address the limitations in commercial PDRN production. We have replaced it with a more appropriate reference that clearly describes the constraints in sourcing and purifying PDRN, particularly in the abstract and introduction (now cited as ref. 31).

Comment) Lines 196-197 – there is here a sentence which is in bold, with no apparent reason; I suppose this is a typo.

Answer) We thank the reviewer for identifying this formatting issue. The bold text was unintended and has been corrected in the revised manuscript.

Comment) Line 220 – There is an improper space after sentence (B); I assume this is an error of text formatting.

Answer) Thank you for noting this formatting error. It has been corrected.

Comment) Line 354 – I think you should add a reference here to document the M1 polarisation consequences.

 Answer) Thank you for the suggestion. We have added relevant references (ref. 42 and 43) to support the statements related to M1 macrophage polarization.

General Comments

Comment) I think the paper could benefit from a dedicated Conclusions section – perhaps you can consider making the last paragraph (lines 363-370) a Conclusions sections, after adding a few more lines summarising your research?

Answer) We appreciate the suggestion. We have revised the manuscript to include a separate Conclusion section, which summarizes the key findings and discusses their implications.

Comment)You mention a lot about the inflammatory pathways (e.g., NF-κΒ, COX-2, etc); I think the manuscript would benefit from a comparison of the effects described in your text with those of other natural products, such as pinosylvin or curcumin (indicative references for these are “Pinosylvin: A Multifunctional Stilbenoid with Antimicrobial, Antioxidant, and Anti-Inflammatory Potential’’ and “Curcumin Attenuates Acrolein-induced COX-2 Expression and Prostaglandin Production in Human Umbilical Vein Endothelial Cells’’ – these are just indicative to orient you towards the specific aspect mentioned; there is not need to cite these papers. You may choose whichever you think are better for your purposes).

Answer) Thank you for this thoughtful comment. In the revised Discussion, we now compare the anti-inflammatory effects of OSB with those of well-known natural products such as curcumin, resveratrol, and quercetin (ref. 39–41), thereby contextualizing our findings within the broader field of natural product-based therapeutics.

Comment) Perhaps you can consider mentioned potential applications of your findings in real clinical situations, for example in viral or bacterial infections; also, I think it would be of benefit to provide some directions for future research.

Answer) We fully agree with the reviewer’s comment that, since our findings are limited to in vitro experiments, additional studies are necessary to confirm their relevance in clinical settings. In this study, we induced inflammatory responses in macrophages using LPS, a component of the Gram-negative bacterial cell wall, which serves as a model for bacterial infection in vitro. For future research, it will be important to validate the therapeutic effects of OSB on inflammation using animal models, including viral and bacterial infection models. In response to the reviewer’s suggestion, we have highlighted the need for further in vivo studies in both the Discussion and Conclusion sections.